# A Biomimetic Flapping Mechanism for Insect Robots Driven by Indirect Flight Muscles

**DOI:** 10.3390/biomimetics10050300

**Published:** 2025-05-08

**Authors:** Yuma Shiokawa, Renke Liu, Hideyuki Sawada

**Affiliations:** 1Department of Applied Physics, School of Advanced Science and Engineering, Waseda University, 3-4-1 Okubo, Shinjuku-ku, Tokyo 169-8555, Japan; askar_liu@fuji.waseda.jp; 2Faculty of Science and Engineering, Waseda University, 3-4-1 Okubo, Shinjuku-ku, Tokyo 169-8555, Japan; sawada@waseda.jp

**Keywords:** flapping mechanism, insect robot, indirect flight muscles, SMA wire

## Abstract

Insect flight mechanisms are highly efficient and involve complex hinge structures that facilitate amplified wing movement through thoracic deformation. However, in the field of flapping-wing robots, the replication of thoracic skeletal structures has received little attention. In this study, we propose and compare two different hinge models inspired by insect flight: an elastic hinge model (EHM) and an axle hinge model (AHM). Both models were fabricated using 3D printing technology using PLA material. The EHM incorporates flexible structures in both the hinge and lateral scutum regions, allowing for deformation-driven wing motion. In contrast, the AHM employs metal pins in the hinge region to reproduce joint-like articulation, while still permitting elastic deformation in the lateral scutum. To evaluate their performance, we employed an SMA actuator to generate flapping motion, and measured the wing displacement, flapping frequency, and exoskeletal deformation. The experimental results demonstrate that the EHM achieves wing flapping through overall structural flexibility, whereas the AHM provides more defined hinge motion while maintaining exoskeletal elasticity. These findings contribute to our understanding of the role of hinge mechanics in bioinspired flapping-wing robots. Future research will focus on optimizing these mechanisms for higher frequency operation, weight reduction, and better energy efficiency.

## 1. Introduction

In recent years, small flying robots have received increasing attention because of their potential to be used in fields such as disaster response, environmental monitoring, and search-and-rescue operations [1]. These robots are especially useful because they can take off and land in narrow spaces, move through complex environments, and quickly provide important information about the situation [2]. This makes them suitable for missions where people or large machines cannot work effectively [3]. Among these small robots, insect-sized robots—machines that imitate the size and movement of real insects—are seen as especially promising [4]. Their compact and lightweight designs allow them to move quickly and efficiently, similarly to insects [5].

Insects, especially those that use indirect flight muscles, can flap their wings rapidly and efficiently [6]. This is made possible by a combination of indirect flight muscles and a specially structured thorax (the middle part of the insect’s body) [7]. Unlike direct flight muscles, which are attached directly to the wings, as seen in insects like dragonflies, indirect flight muscles (common in flies and bees) are not connected directly to the wings [8]. Instead, these muscles cause deformation of the thorax exoskeleton [9]. This movement is transmitted to the wings’ flapping through a complex hinge mechanism. This system also stores energy passively and allows for fast wingbeats with low energy input [10].

Many studies have tried to reproduce insect flight in robots [11]. For example, the RoboBee X-Wing, developed at Harvard University, uses piezoelectric actuators to flap its wings at up to 200 Hz [12,13], and another robot from MIT uses dielectric elastomer actuators (DEAs) to flap at 400 Hz [14]. These robots are major achievements, but they work differently from real insects. In most cases, the wing motion is created by directly vibrating the actuators, not through a chain of movement involving the thorax, hinge, and wing. As a result, their energy efficiency and similarity to real insect mechanics are still limited.

Another problem is that many of these systems need high-voltage power supplies, often more than 200 volts, which makes it difficult to build lightweight, autonomous flying robots [15]. Even though materials and electronic components have improved, it is still challenging to fully reproduce the energy-efficient flight system of real insects—especially the function of the thorax and hinge structure.

Meanwhile, anatomical research has been conducted to understand the structure and function of insect thoraxes, especially in flies (Diptera). Past models—such as the “click mechanism” by Boettiger and Furshpan [16], the hinge motion study by Miyan and Ewing [17,18], and the more recent work by Ennos [19]—have helped to explain how thorax movement and hinge structures work together to flap the wings. Ennos, in particular, pointed out that deformation in the lateral scutum (side part of the thorax) plays an important role in moving the wings through small joint-like parts called axillary sclerites [20]. These biological insights can be used to design better robot hinges.

Shape-memory alloy (SMA) wires are used as actuators that contract when heated by an electric current and return to their original length when cooled [21]. This simple heating and cooling cycle allows them to generate a strong pulling force along the contracting direction, with a fast movement [22]. Because of these features, SMA wires have been used in various applications [23]. For instance, SMA wires have been utilized in inchworm-type robots, which exploit the shape-memory effect to mimic the characteristic crawling motion through temperature-induced shape changes [24]. In another example, SMA wires are used in tactile feedback devices by applying periodic pulse currents to generate high-frequency vibrations [25]. Additionally, their ability to create a strong force upon heating makes them ideal for use as artificial muscles, as seen in robotic hands that closely replicate human finger movements [26].

In this research, we present and compare two types of flapping-wing robots inspired by the flight mechanics of flies. The first model, called an elastic hinge model (EHM), uses flexible 3D-printed parts that bend to produce wing motion. The second model, called an axle hinge model (AHM), also uses 3D-printed parts and additional small metal pins to make clearer, joint-like wing movement. Both robots employ the same SMA wire actuators, which mimic the muscle movement of insects. The wings are made the same in both models so that we can focus only on how the hinge type affects performance.

We compare these two models by measuring how far the wings flap, how fast they can flap, and how much the thorax parts deform during motion. We also compare the results with real insect flight data. Our findings can help to improve the design of flapping-wing robots, especially in making them more energy-efficient. This study contributes to the field of biomimetic robotics by providing experimental evidence on how thorax-driven flapping can be implemented in robotic systems.

## 2. Biological Structure of Insect Wings

### 2.1. Mechanism of Flapping Generation in Insects

As shown in Figure 1, insect wing flapping is primarily driven by deformation of the thoracic exoskeleton, which results from the alternating contraction of the dorso-longitudinal muscles (DLMs) and the dorso-ventral muscles (DVMs) [7]. This cyclic muscular activity induces structural deformation of the thorax, thereby producing the up-and-down motion of the wings. The actuation process follows a hierarchical chain of mechanical transmission: muscular contraction → thoracic deformation → hinge structures → wing movement.

### 2.2. Types of Flight Muscles

Insect flight muscles are typically classified into two main categories: direct flight muscles and indirect flight muscles. As shown in Figure 2, insects with direct flight muscles, such as dragonflies, generate wing movement through muscles that are directly attached to the wing base, producing motion via direct contraction. In contrast, insects with indirect flight muscles, such as flies and bees, have muscles that are attached to the thoracic exoskeleton, rather than the wings themselves. Wing movement is achieved indirectly, as muscular contractions induce deformation of the thorax, which is then transmitted to the wings via specialized hinge structures. Indirect flight systems generally support higher wingbeat frequencies and improved energy efficiency, due to the elastic energy storage and recovery enabled by the thoracic architecture.

### 2.3. Thoracic Deformation and Elasticity

In insects that utilize indirect flight muscles, deformation of the thoracic exoskeleton plays a fundamental role in generating wing motion [19]. The exoskeleton exhibits inherent elastic properties, enabling it to store mechanical energy during muscle contraction and release it during relaxation [27]. This cyclic storage and release of energy significantly contributes to efficient high-frequency wing flapping with minimal energy dissipation. In addition, the system can keep flapping on its own using repeated muscle contractions, which helps insects to fly with high energy efficiency.

### 2.4. Role and Complexity of Hinge Structures

The deformation of the thoracic exoskeleton must be transmitted to the wings through specialized hinge structures located at the wing base. These hinges are made up of multiple sclerites and joints, forming a highly complex system that allows movement in multiple directions [7]. Although many anatomical and biomechanical studies have been conducted, the detailed motion of each component during wing flapping is still not fully understood [19]. Because of this, many theoretical models have been proposed to explain how these hinge structures work and how they contribute to wing movement.

### 2.5. Ennos Model of Hinge Structure

Among the proposed hinge models, the Ennos model is one of the most widely referenced [15]. It explains how wing motion is generated through coordinated deformation of the thoracic exoskeleton, particularly involving the scutum and lateral scutum. Figure 3 shows a schematic diagram of the Ennos hinge model, illustrating how wing motion arises from thoracic deformation. In this study, we design our robotic hinge structures based on the principles of the Ennos model, aiming to replicate the mechanical behavior of insect flapping as closely as possible.

## 3. Structural Design and SMA-Based Actuation of Hinge Models

### 3.1. Shared Design Elements

Both the EHM and the AHM share the same overall structure, wing shape, and actuation system to ensure that the effects of hinge mechanics can be isolated and fairly compared. All structural components were fabricated using fused deposition modeling (FDM) 3D printing with polylactic acid (PLA), a commonly used thermoplastic known for its moderate stiffness and ease of processing.

The wings used in both models were identical in geometry and mass, and were constructed using PLA for the leading edge and a thin polyester film as the membrane. The mass of each wing was approximately 1.2 g. A 3D projection view of the wing vein is shown in Figure 4a. To replicate the function of insect indirect flight muscles, a shape-memory alloy (SMA) wire actuator was installed in each thoracic frame. The SMA wire contracts when heated by an electric current, causing thoracic deformation that mimics the contraction of dorso-ventral muscles. The return motion is facilitated by the elastic recovery of the structure itself.

By holding constant the wing structure and actuator, this design ensured that any observed differences in flapping performance could be attributed solely to the mechanical differences in the hinge region.

### 3.2. Elastic Hinge Model (EHM)

The EHM was designed to reproduce the compliant behavior of insect thoracic hinges using only flexible materials. The hinge and lateral scutum regions were constructed as a continuous, monolithic structure and entirely fabricated from PLA using 3D printing. The mass of the thoracic structure (EHM) was 13.6 g. When the SMA actuator contracts, these areas undergo elastic deformation, resulting in thoracic compression that drives the flapping motion of the wings. The 3D projection view of the EHM is shown in Figure 4b.

This model does not include any discrete articulation components, such as pins or joints. Instead, the wing motion relies entirely on the flexibility of the material and the geometry of the hinge region. As a result, the hinge motion is passive and spatially distributed. This design allows for simple fabrication and high compliance, although the material flexibility may absorb actuator force and introduce damping effects, which can reduce wingbeat amplitude, particularly at higher actuation frequencies.

Overall, this model represents a minimalistic bioinspired hinge, where all actuation forces are transmitted exclusively through elastic deformation.

### 3.3. Axle Hinge Model (AHM)

The AHM incorporates metal pins to more closely replicate the articulated joint mechanisms found in biological hinge structures. In this design, the hinge region consisted of separately printed thoracic components with integrated pinholes. Steel pins were inserted to form a simple revolute joint, allowing localized and controlled rotation around the hinge axis. The mass of the thoracic structure (AHM) was 13.7 g. The 3D projection view of the EHM is shown in Figure 4c.

The lateral scutum is also made from PLA, and maintains a moderate degree of flexibility. During actuation, it deforms slightly, similarly to natural insect thoraxes. However, the axle structure concentrates the hinge movement and reduces unwanted deformation in surrounding thoracic areas.

This model was developed to evaluate whether the addition of a clearly defined rotational axis enhances wing displacement and motion clarity during actuation. Furthermore, the configuration more closely reflects the function of axillary sclerites, as described in insect thoracic anatomy and modeled by Ennos.

As shown in Figure 5, the hinge model, wings, and SMA wire actuator were assembled, and a rear support component was attached to stabilize and synchronize the motion of the left and right wings. The assembled structure was fixed onto a base for the flapping experiments. The wingspan of the assembled model was approximately 370 mm. Including the supporter and the base, the total mass was 57.9 g.

### 3.4. Generation of Flapping Motion Using Shape-Memory Alloy (SMA) Actuator

Shape-memory alloys (SMAs) are a class of shape-memory materials that can recover their original shape after undergoing quasi-plastic deformation [28]. This shape-memory effect occurs due to a solid-state phase transformation triggered by temperature changes. As illustrated in Figure 6, SMAs mainly exist in two phases [29]. The martensite phase is stable at lower temperatures and is relatively soft, allowing deformation under external stress. In contrast, the austenite phase is stable at higher temperatures and resists deformation. When an SMA is deformed in the martensite state and subsequently heated beyond a critical transformation temperature, it shifts into the austenite phase and recovers its original shape. This phenomenon is referred to as the shape-memory effect.

Thin SMA wires exhibit rapid thermal response due to their high surface-area-to-volume ratio [30]. When an electric current passes through the wire, Joule heating increases the temperature, causing a phase transformation and resulting in an instantaneous contraction of the wire [31]. Once the current is turned off, the wire cools down through heat radiation in the air, and returns to its original length. Because of their fast thermal cycle, thin SMA wires can vibrate when driven by repeated pulse currents. As shown in Figure 7, a pulse current is characterized by three independent parameters: amplitude, frequency, and duty ratio (the ratio of the pulse width to the period). By adjusting these parameters, the vibration behavior of the SMA wire can be controlled. Thus, the SMA wire can be considered as a good actuator for applications with small spatial availability and weight limitations.

## 4. Evaluation and Results

### 4.1. Experimental Setup

To evaluate the mechanical behavior of the two hinge models, a test platform was developed to measure wing displacement, flapping frequency, and thoracic deformation. The overall setup is shown in Figure 8a. Both the EHM and the AHM were mounted on this platform under identical conditions to ensure fair comparison.

Flapping motion was generated using a shape-memory alloy (SMA) wire integrated into the thoracic structure of each model. The SMA wire was driven by DC power supply controlled by a pulse width modulation (PWM).

To quantify both wingbeat amplitude and thoracic deformation, white markers (2 mm in diameter) were placed at specific locations on each model. Five markers were attached at equal intervals along the wing vein to track wing motion, and three additional markers were applied to the thoracic region: one on the upper scutum, one at the junction between the scutum and the lateral scutum, and one on the lateral scutum itself.

The models were actuated and recorded from a lateral view using a high-speed camera. Using OpenCV, the coordinates of all markers were extracted for each video frame. The wingbeat amplitude was calculated as the maximum positional change of the wing vein markers during one flapping cycle (as shown in Figure 8b). Thoracic deformation was quantified by computing the angular variation formed by the three thoracic markers, representing local bending of the lateral scutum area during actuation (as shown in Figure 8c).

Figure 9 shows a circuit diagram of the control system used to drive the SMA actuator in the robot. An Arduino Mega microcontroller is used to generate the control signals, and a transistor (2SD880) is employed in the circuit. The PWM signal from the Arduino controls the switching behavior of the transistor. This switching action determines the frequency and duty ratio of the pulse current that flows through the SMA wire. The system is powered by a DC voltage source set to 30 V, which is sufficient to generate the required Joule heating for SMA contraction. The PWM signal is configured with a frequency range from 0.6 Hz to 3.0 Hz and a duty ratio between 2% and 11%, depending on the test condition.

The upper limit of 3.0 Hz was set because, due to the greater wing inertia resulting from the model’s scaled-up size, several tens of times larger than that of real insects, realistic flapping motions could not be sufficiently replicated at higher frequencies. Additionally, the duty ratio was adjusted for each frequency, because as the frequency increased, the duration of each cycle shortened, making it necessary to provide a longer relative power-on time to ensure adequate heating and contraction of the SMA wire. These parameters were adjusted experimentally to optimize the amplitude and responsiveness of the flapping motion.

By setting the actuation frequency at several Hz and adjusting the duty ratio accordingly, characteristic flapping behavior around the hinge structure, such as wing rotational motion, was clearly observed. This setting enabled us to analyze the functions of indirect flight muscles and the structures of the hinge in more detail.

Figure 10 shows the flapping motion generated by the SMA actuator. The images above the timeline illustrate the side view, while those below the line show the front view of the model. The driving conditions were set to a frequency of 3.0 Hz and a duty ratio of 9%. Within the time shown in the figure, the pulse current flows through the SMA wire for 0.03 s from t = 0.17 [s]. As observed in the figure, the wing moves upward when the SMA wire contracts due to Joule heating.

Furthermore, during the upstroke, a twisting of the wing surface, which is called supination, can be seen. This twisting motion is important in real insect flight [32], and was successfully reproduced in our model as a result of passive deformation of the wing structure during flapping.

### 4.2. Experiment of Elastic Hinge Model

This section details the flapping performance of the EHM under different actuation frequencies and PWM duty ratios. For each condition, the wingbeat amplitude at the wingtip was measured in both linear displacement (mm) and angular rotation (°). The results are summarized in Table 1.

At 0.6 Hz, the wingbeat amplitude clearly increases as the duty ratio rises, reaching 46.0 mm and 20.1° at 6% duty. A similar trend appears at 1.0 Hz, where the amplitude reaches 44.1 mm and 19.4° at 9% duty. At 2.0 Hz, the amplitude remains relatively high, reaching 40.6 mm and 18.1° at 10% duty. However, at 3.0 Hz, the overall performance drops, and the maximum amplitude at 10% duty is 32.2 mm and 14.8°.

These results show that lower actuation frequencies and higher duty ratios lead to larger wingbeat amplitudes. The highest amplitude recorded in all tests is 46.0 mm and 20.1°, achieved at 0.6 Hz with a 6% duty ratio. This represents the best performance of the EHM.

On the other hand, when the actuation frequency becomes higher, the wingbeat amplitude clearly decreases. This seems to happen because the wing does not have enough time to return to its original position before the next upstroke starts. Figure 11 shows the horizontal displacement of the wing vein markers. At 0.6 Hz (Figure 11a), the wingtip reaches about −35 mm, but at 3.0 Hz (Figure 11b), it only reaches about −20 mm. This means that the SMA wire starts reheating and contracting before it fully cools and extends, and before the structure elastically returns. Therefore, the short recovery time at higher frequencies likely causes the decrease in wingbeat amplitude.

The thoracic deformation analysis showed that the lateral scutum bent inward during the wing upstroke. As shown in Figure 12, the angle between the scutum and lateral scutum decreased during this motion, indicating localized inward bending. This result was obtained under the actuation condition of 0.6 Hz and a 6% duty cycle, which corresponds to the peak performance observed in Table 1. This deformation occurred in synchrony with thoracic compression and wing elevation, suggesting that the flapping motion was generated through deformation of the exoskeletal structure. These observations confirm that flapping can be achieved by combining SMA-induced contraction with structural elasticity and hinge geometry. This behavior closely matches a key feature of Ennos’ hinge model, indicating that the current design successfully replicates an important aspect of the biological flapping mechanism.

### 4.3. Experiment of Axle Hinge Model

This section presents the flapping performance of the AHM under various actuation frequencies and PWM duty ratios. All experiments were conducted under the same conditions as those used for the EHM, for direct comparison. The driving voltage was fixed at 30 V, and the actuation frequencies and duty ratios were matched to those of the EHM.

For each condition, the wingbeat amplitude at the wingtip was measured in both linear displacement (mm) and angular rotation (°). The measured results are summarized in Table 2.

At 0.6 Hz, the wingbeat amplitude increases significantly with higher duty ratios, reaching a maximum of 134.6 mm and 57.9° at a 5% duty ratio. At 1.0 Hz, the amplitude reaches 123.1 mm and 50.1° at 7%, with moderate variation across neighboring conditions. At 2.0 Hz, the model maintains relatively high performance, achieving 97.8 mm and 39.3° at 10% duty. However, at 3.0 Hz, the amplitude declines substantially, with a maximum of 61.7 mm and 25.3° observed at 9%, and further increases in duty ratio do not improve performance.

A general trend is observed whereby lower actuation frequencies combined with higher duty ratios lead to greater wingbeat amplitudes. Across all tests, the maximum amplitude is recorded at 0.6 Hz with a 5% duty ratio, reaching 134.6 mm and 57.9°, which represents the peak performance of the AHM.

At higher actuation frequencies, the AHM also exhibits a reduction in wingbeat amplitude. This is evident from the horizontal displacement of the wing vein markers shown in Figure 13. In Figure 13a, at 0.6 Hz, the wingtip reaches a minimum displacement of approximately −58 mm, whereas in Figure 13b, at 3.0 Hz, it only reaches about −50 mm. This indicates that the next flapping motion begins before the SMA wire has fully cooled and re-extended, and before the hinge structure has returned to its neutral position. These findings suggest that, despite having a well-defined rotational axis, the AHM also faces limitations in elastic recovery at higher frequencies, which contributes to the reduction in flapping amplitude.

The thoracic deformation analysis of the AHM showed that the lateral scutum bent inward during the wing upstroke. This movement occurred at the same time as the thorax was compressed and the wing was lifted. This confirms that flapping motion can be generated through deformation of the exoskeleton caused by SMA actuation. Figure 14 shows that the angle between the scutum and lateral scutum decreased during upstroke, indicating localized bending. This result was obtained under the actuation condition of 0.6 Hz and a 5% duty cycle, which corresponds to the peak performance observed in Table 2. This result suggests that the model replicates a key behavior seen in insects with indirect flight muscles. The flapping motion was achieved by combining material flexibility, structural elasticity, SMA contraction, and a hinge design inspired by the insect thorax. These findings closely match the behavior described in Ennos’ hinge model.

### 4.4. Comparison and Discussion

The results obtained from the two hinge models reveal clear differences in flapping performance and structural characteristics. The AHM consistently achieved larger wingbeat amplitudes than the EHM across all tested conditions. For instance, at 0.6 Hz, the axle model reached a maximum of 134.6 mm and 57.9°, while the elastic model peaked at 46.0 mm and 20.1°. A summary graph of the maximum angular displacements for both models under all measurement conditions is presented in Figure 15. These results suggest that the presence of a defined rotational axis in the axle design facilitates more effective transmission of actuator force to the wings.

The EHM, by contrast, offers a simpler and more compliant structure. This design may provide advantages in terms of fabrication ease, structural integration, and potential energy storage through distributed elasticity. However, its flapping performance is generally lower. While the axle model outperformed the elastic model in absolute amplitude, both exhibited a noticeable decline in performance at higher actuation frequencies. These observations suggest that the reduction in flapping amplitude at higher frequencies in both models was primarily due to insufficient recovery time between cycles, resulting from delayed SMA wire cooling and limited structural elasticity.

Deformation of the lateral scutum was observed in both models, occurring in synchrony with the wing upstroke. This behavior aligns with the hinge mechanism described in Ennos’ model, and indicates that the proposed hinge structures can replicate important features of insect thoracic dynamics.

In this study, the wing elevation was generated by the contraction of the dorso-ventral muscles, which also caused supination due to the elasticity of the wing veins. However, the wing depression was reproduced using only the elasticity of the exoskeleton, which resulted in unclear or absent pronation. In real insects, wing depression is driven by the contraction of dorso-longitudinal muscles, which are arranged orthogonally to the dorso-ventral muscles. Therefore, the current model does not fully reproduce the complete wingbeat cycle. In addition, real insects have steering muscles located near the wing base. These muscles allow for fine control of wing motion, such as generating figure-of-eight flapping trajectories [33,34]. Since such muscles were not included in this model, precise control of wing movement was not possible. In addition, it has been proposed that insects may use shifts in their body mass or center of gravity as part of their flight control strategies [35]. This suggests that insect flight control may rely not only on active muscle actuation near the wings, but also on body-level dynamics.

In future research, we plan to extend the thoracic structure along the longitudinal axis and add an actuator to replicate the function of the dorso-longitudinal muscles. This will help us to better reproduce the wing depression and pronation, making the wingbeat motion more similar to that of real insects. We also plan to implement a fine control system inspired by the steering muscles in insects. This will enable the robot to perform asymmetric wing motions and improve its postural control. These improvements will contribute to the development of more advanced insect-like flapping-wing robots.

In summary, the AHM demonstrates superior wingbeat amplitude and motion clarity, due to its articulated structure, while the EHM features a lightweight and compliant design with fewer mechanical components. The choice between these two designs may depend on the specific application, and future designs may benefit from combining structural compliance with defined articulation to achieve both efficiency and simplicity. Future improvements should also include active downstroke and steering control for better flight.

## 5. Conclusions

In this study, we developed and tested two flapping-wing robot models inspired by insect thoracic hinge mechanisms: the elastic hinge model (EHM) and the axle hinge model (AHM). Both models were made using 3D printing with PLA, and were actuated by SMA wires to simulate muscle contraction. They were evaluated under the same conditions to allow fair comparison.

The AHM showed higher wingbeat amplitudes than the EHM in all tested cases. This result is likely due to the presence of a defined rotational axis, which enabled more efficient force transmission. On the other hand, the EHM had a simpler, one-piece structure with fewer mechanical parts, and was easier to build. It also offered more structural flexibility. However, both models showed reduced performance at higher frequencies, mainly because the SMA wire did not have enough time to cool and return to its original length. The elastic structure also could not fully recover between flapping cycles.

In both models, the lateral scutum was observed to bend inward during the wing upstroke. This deformation is consistent with the hinge mechanism described by previous research, suggesting that both designs can replicate key features of real insect flight.

In the present study, the models focused on planar thoracic deformation, primarily replicating the motion driven by dorso-ventral muscles. In future work, we plan to extend the design along the longitudinal axis to incorporate the function of dorso-longitudinal muscles. This modification will allow the system to reproduce the downward flapping motion caused by longitudinal muscle contraction and enable a more faithful replication of thoracic elasticity.

## Figures and Tables

**Figure 1 biomimetics-10-00300-f001:**
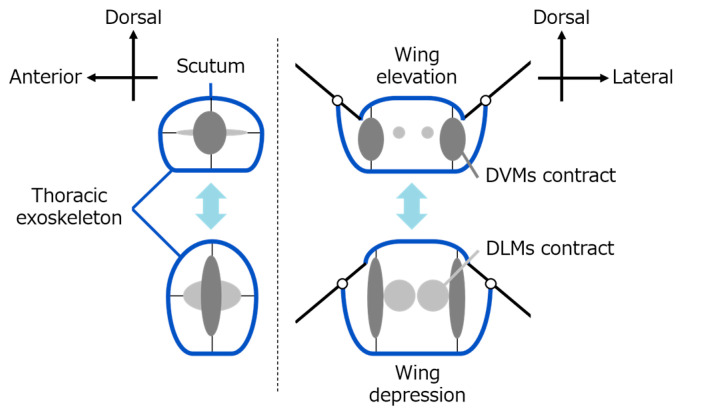
Indirect flight muscles and thoracic deformation in insect wing flapping.

**Figure 2 biomimetics-10-00300-f002:**
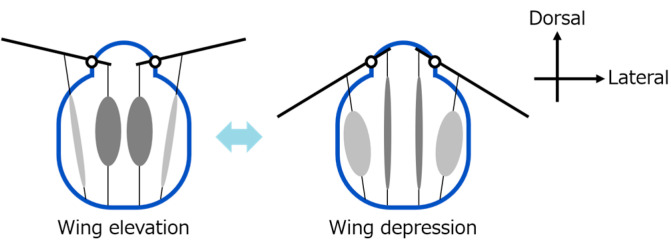
Direct flight muscles in insect wing flapping.

**Figure 3 biomimetics-10-00300-f003:**
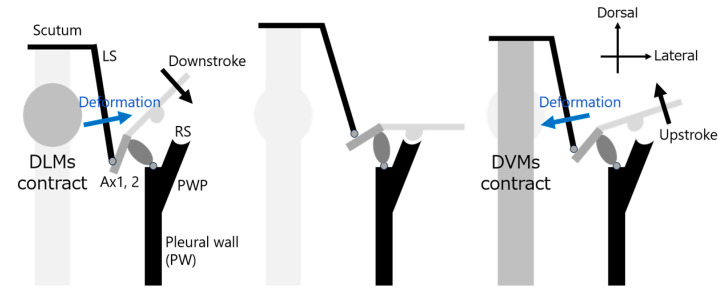
Schematic of Ennos hinge model and thoracic deformation.

**Figure 4 biomimetics-10-00300-f004:**
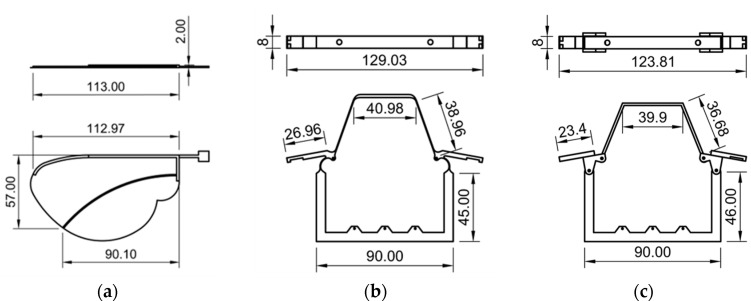
The 3D-printed models: (**a**) the wing vein; (**b**) the EHM structure; (**c**) the AHM structure.

**Figure 5 biomimetics-10-00300-f005:**
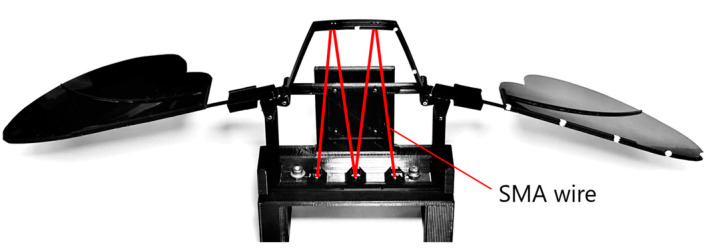
Assembled view of 3D-printed flapping mechanism.

**Figure 6 biomimetics-10-00300-f006:**
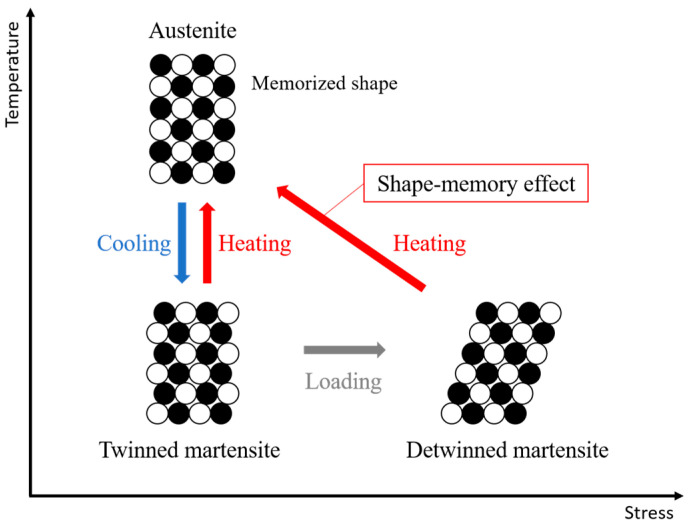
Schematic diagram of phase transformation in shape-memory alloys under temperature and stress.

**Figure 7 biomimetics-10-00300-f007:**
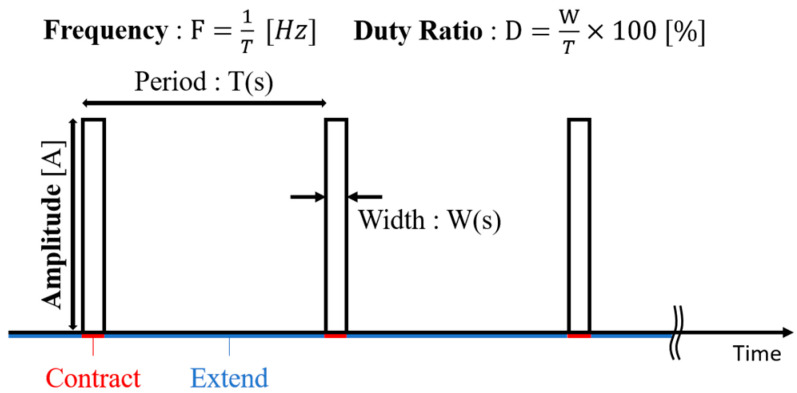
PWM control for SMA vibration.

**Figure 8 biomimetics-10-00300-f008:**
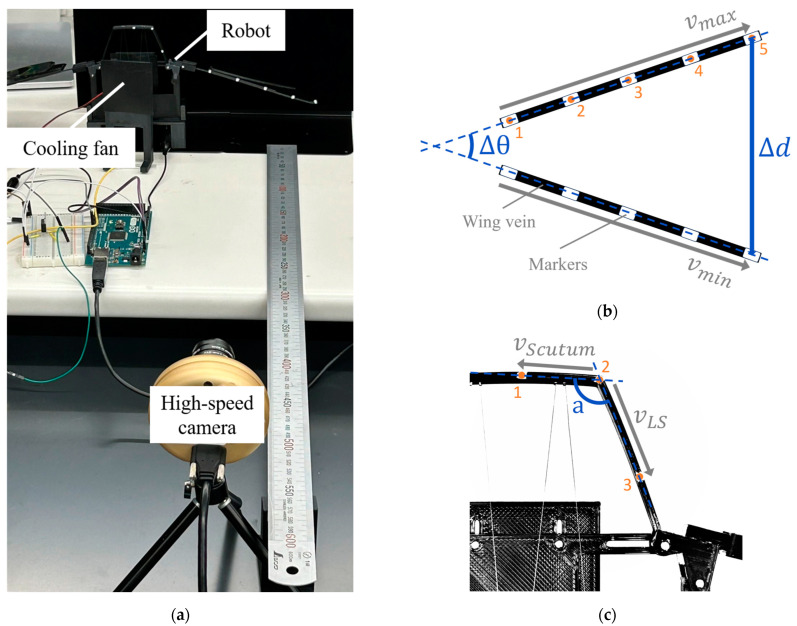
Experimental setup and measurement methods for flapping kinematics and thoracic deformation: (**a**) experimental setup for lateral-view high-speed imaging; (**b**) tracking method for wingbeat amplitude and maximum angular displacement using wing vein markers; (**c**) measurement method for lateral scutum deformation using thoracic markers.

**Figure 9 biomimetics-10-00300-f009:**
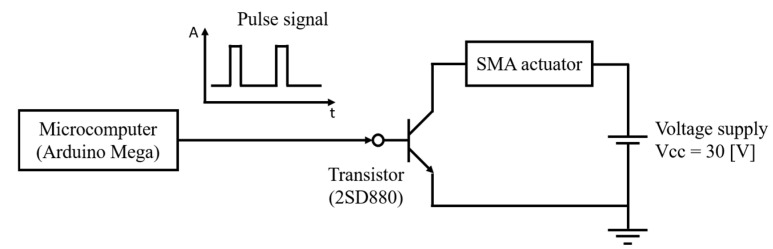
Circuit diagram of control system for driving SMA actuator.

**Figure 10 biomimetics-10-00300-f010:**
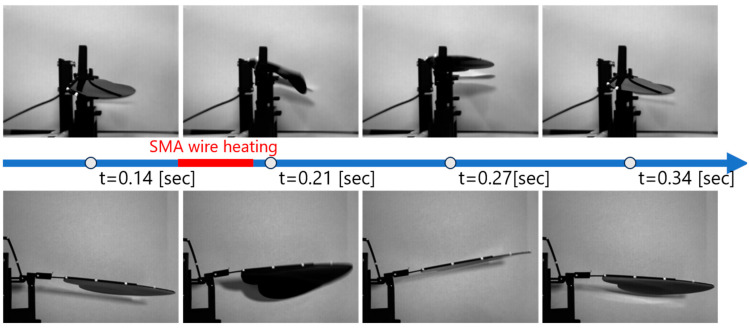
Flapping motion captured from side and front views.

**Figure 11 biomimetics-10-00300-f011:**
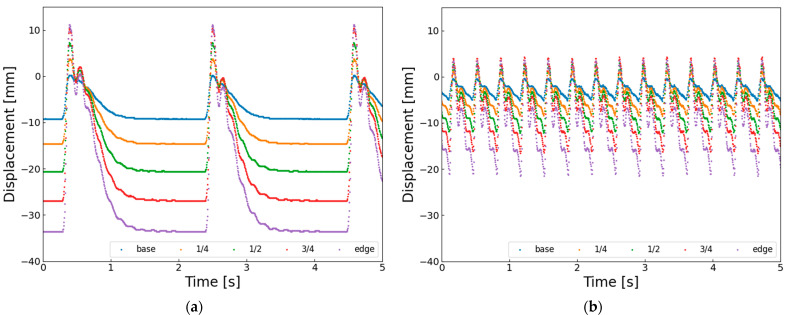
Horizontal displacement of five wing vein markers in elastic hinge model (EHM) under different driving conditions: (**a**) EHM flapping at 0.6 Hz with 6% duty ratio; (**b**) EHM flapping at 3.0 Hz with 11% duty cycle.

**Figure 12 biomimetics-10-00300-f012:**
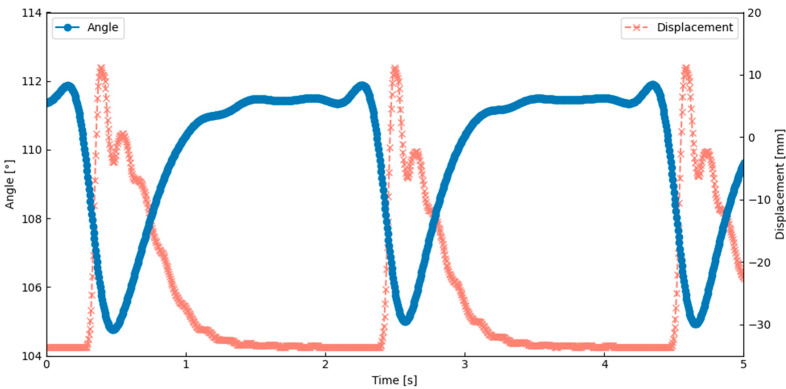
The relationship between wing motion and thoracic deformation in the elastic hinge model (EHM).

**Figure 13 biomimetics-10-00300-f013:**
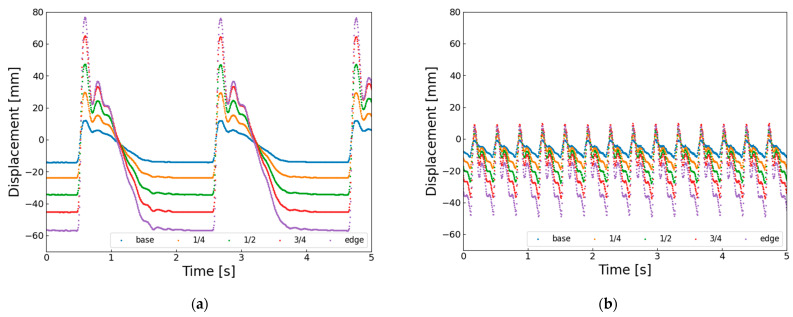
Horizontal displacement of five wing vein markers in axle hinge model (AHM) under different driving conditions: (**a**) AHM flapping at 0.6 Hz with 6% duty ratio; (**b**) AHM flapping at 3.0 Hz with 11% duty cycle.

**Figure 14 biomimetics-10-00300-f014:**
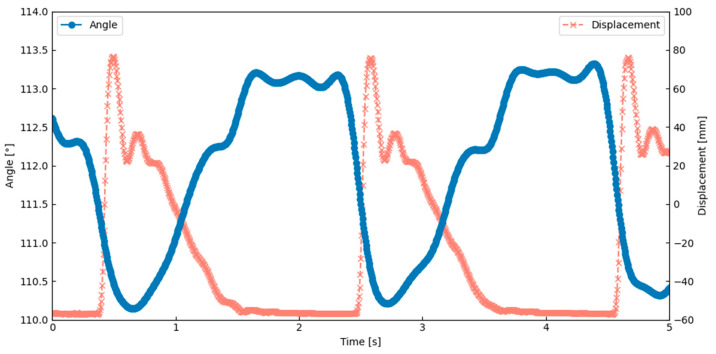
The relationship between wing motion and thoracic deformation in the axle hinge model (AHM).

**Figure 15 biomimetics-10-00300-f015:**
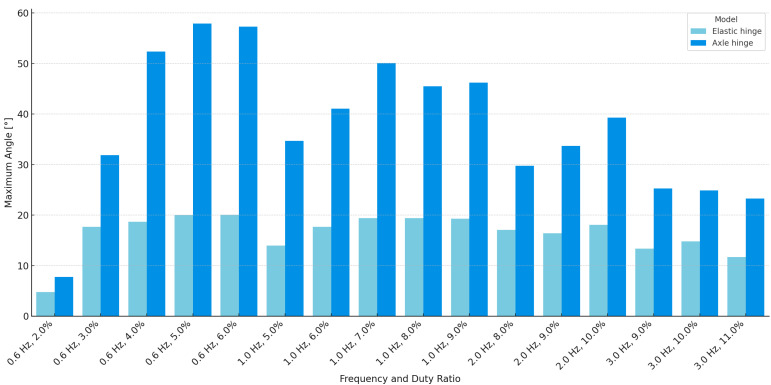
Comparison of maximum flapping angles in EHM and AHM under various conditions.

**Table 1 biomimetics-10-00300-t001:** Amplitude and maximum angle measurements in EHM experiments.

Frequency [Hz]	Duty Ratio [%]	Amplitude (Δd) [mm]	Maximum Angle (Δθ) [°]
0.6	2	10.6	4.8
	3	39.9	17.7
	4	42.2	18.7
	5	45.9	20.0
	6	46.0	20.1
	7	-	-
1.0	5	31.3	14.0
	6	39.9	17.7
	7	44.1	19.4
	8	44.1	19.4
	9	44.0	19.3
	10	-	-
2.0	8	38.1	17.1
	9	36.4	16.4
	10	40.6	18.1
	11	-	-
3.0	9	29.1	13.4
	10	32.2	14.8
	11	25.2	11.7
	12	-	-

**Table 2 biomimetics-10-00300-t002:** Amplitude and maximum angle measurements in AHM experiments.

Frequency [Hz]	Duty Ratio [%]	Amplitude (Δd) [mm]	Maximum Angle (Δθ) [°]
0.6	2	17.2	7.8
	3	75.2	31.9
	4	123.6	52.4
	5	134.6	57.9
	6	133.2	57.3
	7	-	-
1.0	5	85.3	34.7
	6	101.1	41.1
	7	123.1	50.1
	8	111.5	45.5
	9	121.4	46.2
	10	-	-
2.0	8	74.0	29.8
	9	84.1	33.7
	10	97.8	39.3
	11	-	-
3.0	9	61.7	25.3
	10	61.2	24.9
	11	57.0	23.3
	12	-	-

## Data Availability

Data can be obtained from the authors upon request.

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
