# Peer review of "A Biomimetic Flapping Mechanism for Insect Robots Driven by Indirect Flight Muscles"

_biomimetics, 2025, doi:10.3390/biomimetics10050300_

Round 1

Reviewer 1 Report

Comments and Suggestions for Authors

This article has certain research significance, but the author's research depth is not sufficient, and the lift data of flapping wings has not been provided yet. Multiple device schematics are for popular science. Thus, I think this paper is not suitable for publication.

1.Please give the reason that limits the frequency to 3Hz.

  1. The weight of each part of the flapping wing mechanism is not given.
  2. The duty ratio input is set as different values under different frequency. Please give the reason for this selection. For instance, when frequency is 0.6, duty ratio is 2,3,4,5,6 and 7%. However, when the frequency is 1.0, the duty ratio is 5,6,7,8,9 and 10%.
  3. Please clarify that the figure 11(or 13) gives the results for which case in Table 1(or 2)? From Figure 11, the angle amplitude s 7 degree while the displacement amplitude is 46 mm.
  4. From Figure 14, the authors may give some lift testing results instead of repeating the similar motion data. Also, it is better to explain in detail how to control the shape memory alloy with the flight control board and the peripheral devices instead of give the PWM control principle in Figure 7.

Reviewer 2 Report

Comments and Suggestions for Authors

Insect flight mechanisms are highly efficient and involve complex hinge structures that facilitate enhanced wing propulsion through thorax deformation. However, most existing flapping-wing robots do not accurately reproduce these hinge mechanisms. In this study, we propose and compare two different hinge models inspired by insect flight: the elastic hinge model (EHM) and the axial hinge model (AHM). Both models were fabricated using 3D printing technology using PLA material. The EHM incorporates flexible structures in both the hinge and side flap regions, allowing for deformation-induced wing motion. In contrast, the AHM uses metal pins in the hinge region to reproduce a joint-like articulation while still allowing elastic deformation in the side flap. To evaluate their performance, we used an SMA actuator to generate flapping motion and measured wing displacement, flapping frequency, and exoskeleton deformation. Experimental results show that EHM achieves wing flapping due to overall structural flexibility, while AHM provides more defined hinge motion while maintaining exoskeleton elasticity. These results provide insights into the role of hinge mechanics in bioinspired flapping wing robots. Future research will focus on optimizing these mechanisms for higher operating frequency, reduced weight, and better energy efficiency.

This paper is devoted to studying the mechanisms of insect wing flapping motion with the aim of applying similar methods to robots inspired by these natural mechanical solutions. The article is very interesting for readers. Based on the analysis of the literature, which describes the investigated real mechanisms of flapping motion in flies and dragonflies, two fundamentally different approaches are presented. One approach is associated with direct muscle action on the wing bases, the other approach is associated with the deformation of a part of the insect's cephalothorax, which leads to wing flaps. Each of these approaches has its own advantages and disadvantages. The authors of the article developed, manufactured and studied using various methods of analysis both variants of the flapping movements of wings for robotics. For this purpose, wings, corresponding hinge systems, housings and actuators were developed and manufactured, providing the conversion of electrical energy into mechanical energy. The article clearly shows the advantages and disadvantages of the two methods when used in the variant for flying robots. In one case, it is possible to provide a greater amplitude of flapping, in the other case, a higher frequency and greater simplicity of the design are ensured. All this is extremely interesting. For some reason, the authors do not discuss the problem that for flight, a simple repetition of wing flapping up and down is not enough. During flight, the wing movement is more complex, it performs movements resembling the figure 8. In addition, the wing must turn itself so that when going up, it does not encounter the same great air resistance as when going down. In some birds, this is ensured by the properties of feathers, which, when air flows over them from above, easily open and let it pass through, and when air flows over them from below, close their hairs and do not let air pass through them. Insects and hymenoptera, as well as hummingbirds, have other methods, and in birds with feathers with such properties, this is not the only way to ensure flight qualities. Rigid wings must turn during flight. That is, more than one degree of freedom is required. Only up and down movement will not ensure flight. In addition, maneuvering also requires the ability to move the wings differently, so that the movements of the left wing differ from the movement of the right wing. It is desirable that further research by the authors of the article also touch upon these aspects of this problem.

Round 2

Reviewer 1 Report

Comments and Suggestions for Authors

The manuscript has been well revised and is suitable for publication.